# Quadrupole topological photonic crystals

Li He[1], Zachariah Addison[1], Eugene J. Mele[1] & Bo Zhen [1✉]

Quadrupole topological phases, exhibiting protected boundary states that are themselves topological insulators of lower dimensions, have recently been of great interest. Extensions of these ideas from current tight binding models to continuum theories for realistic materials require the identification of quantized invariants describing the bulk quadrupole order. Here we identify the analog of quadrupole order in Maxwell's equations for a gyromagnetic photonic crystal (PhC) through a double-band-inversion process. The quadrupole moment is quantized by the simultaneous presence of crystalline symmetry and broken time-reversal symmetry, which is confirmed using three independent methods: analysis of symmetry eigenvalues, numerical calculations of the nested Wannier bands and the expectation value of the quadrupole operator. Furthermore, we reveal the boundary manifestations of quadrupole phases as quantized edge polarizations and fractional corner charges. The latter are the consequence of a filling anomaly of energy bands as first predicted in electronic systems.

[1] Department of Physics and Astronomy, University of Pennsylvania, Philadelphia, PA 19104, USA. ✉email: bozhen@sas.upenn.edu

Symmetries play a pivotal role in understanding and classifying various topological phases of matter[1–4]. In periodic media, systems with different symmetries can admit different classifications characterized by quantized topological invariants. Furthermore, interface states in symmetry-protected topological (SPT) phases can only robustly exist, between two topologically distinct regions, when required bulk symmetries are preserved at the boundaries. In the simplest example of SPT phases, a one-dimensional (1D) Su-Schrieffer-Heeger (SSH) model, the topological invariant—Zak phase[5]—is only quantized when inversion symmetry is preserved, leading to edge states that are protected by non-zero bulk polarizations. Recently, the concept of polarization, or bulk dipole moment in crystals, has been generalized to include multipole moments, such as quadrupole and octupole moments, leading to the discovery of higher-order topological insulators (HOTIs)[6–10]. In particular, these HOTIs have vanishing dipole densities but non-zero higher-order multiple moments, which can be quantized by certain crystalline symmetries, such as reflection and rotation. In contrast to conventional topological insulators (TIs) that support gapless boundary states, HOTIs exhibit protected boundaries that are, themselves, TIs in lower dimensions.

Recently, quadrupole TIs have been demonstrated in a number of classical systems, ranging from microwave[11] and optics[12] to acoustics[13] and circuits[14,15]. Most experiments are analyzed as lattice models, following the tight-binding approximation. In these lattice models, the adopted symmetry required to quantize the quadrupole moment is reflection or fourfold rotation ($C_4$) with threaded $\pi$-flux, and hence these systems are time-reversal invariant. Unfortunately, most realistic systems with subwavelength features, such as photonic crystals, cannot be modeled discretely but require a different continuum approach. For example, the practical implementations of many lattice models no long preserve quantized bulk quadrupole moments in the continuum theory. In addition, most experimental works solely used the existence of in-gap corner states as the measure of bulk quadrupole topology. The validity relies on the presence of additional chiral symmetry in the lattice model, which is often not preserved in the continuum theory.

Here, we find solutions to continuous Maxwell's equations in gyromagnetic photonic crystals that are the electrodynamic analogs to quadrupole topological phases. The proposed topological PhCs have quantized bulk quadrupole moments, which are protected by the simultaneous presence of crystalline symmetries and broken time-reversal symmetry ($T$)[16]. In particular, we show it is essential to break time-reversal symmetry—to open the energy gap—while preserving crystalline symmetries—to quantize bulk dipole and quadrupole moments. In addition, we note the existence of corner states is neither sufficient nor necessary condition for quadrupole phases. Instead, we validate the bulk quadrupole nature through analyzing the Wannier band polarization and its manifestations at boundaries as quantized edge polarizations and fractional corner charges. All calculations are based on realistic parameters that are readily available in the microwave regime[17,18].

## Results

**Quadrupole phase transition through band inversion.** We start by presenting the topological phase transition between a trivial (Fig. 1b) and a quadrupole two-dimensional (2D) PhC (Fig. 1c). The 2D PhC consists of gyromagnetic rods in air and is homogeneous along the out-of-plane ($z$) direction. Experimentally, this boundary condition can also be realized using metals[17,18]. The gapless transition point is achieved in a $2 \times 2$ super-cell structure with four square rods (Fig. 1a). All rods are identical in shape and are of the same gyromagnetic material of Yttrium Iron Garnet

(YIG) with isotropic dielectric permittivity of $\epsilon = 15\epsilon_0$ and inplane permeability $\mu = 14\mu_0$. To break time-reversal symmetry, an external magnetic field is applied along the out-of-plane direction ($z$), which induces complex-valued off-diagonal terms in the permeability tensor of YIG[19]:

$$\bar{\bar{\mu}} = \begin{bmatrix} \mu & i\kappa & 0 \\ -i\kappa & \mu & 0 \\ 0 & 0 & \mu_0 \end{bmatrix} \qquad (1)$$

This gyromagnetic response, $\mu_{xy} = \mu_{yx}^* = i\kappa$, breaks $T$ but preserves $C_4$ and $M_{x(y)}T$. Here $M_{x(y)}$ is mirror reflection that transforms $x(y)$ to $-x(y)$. At the phase transition, all rods are placed at $a/2$ away from their neighbors; the corresponding band structure for TM modes ($E_z$, $H_x$, $H_y$) has twofold (fourfold) degeneracies at the center (corner) of the folded Brilluion zone $\Gamma$ (M). Both degeneracies are lifted when the four rods are simultaneously displaced inward ($d < 0$, Fig. 1b) or outward ($d > 0$, Fig. 1c) along the diagonal lines of the unit cell. On either side of the transition point, the band structure is fully gapped owing to $T$-breaking (shaded in yellow) and supports two topologically distinct phases determined by the displacement $d$: for the choice of cell in Fig. 1, inward displacements with negative $d$ give rise to trivial phases, whereas outward displacements with positive $d$ correspond to quadrupole phases.

Next, we present our calculations of the quadrupole topological invariant $q_{xy}$, using three different approaches, and demonstrate the phase transition between $q_{xy} = 0$ and $q_{xy} = 1/2$ by displacing the dielectric rods. We start by evaluating $q_{xy}$ using the $C_4$ eigenvalues at high-symmetry **k** points of all bands below the gap[6,16]:

$$e^{i2\pi q_{xy}} = r_4^+(\Gamma)r_4^{+*}(M) = r_4^-(\Gamma)r_4^{-*}(M). \qquad (2)$$

Here, $r_4^+$ ($r_4^-$) is the $C_4$ eigenvalue of a mode with $C_2$ eigenvalue $r_2 = +1(-1)$; naturally, $r_4^+ = \pm 1$ and $r_4^- = \pm i$. Accordingly, a quadrupole topological phase transition happens when two pairs of bands switch their $C_4$ eigenvalues at the same time—a process we call "double-band-inversion". Specifically, the double-band inversion happens in our system when $d$ changes from negative to positive: through this process, the TM mode at M with a phase winding of $+2\pi$ in the $E_z$ mode profile ($r_4 = i$, labeled as green) switches position with the one with the winding of $-2\pi$ ($r_4 = -i$, label in red); meanwhile, the two modes with $r_4 = \pm 1$ at M also switch positions. On the other hand, all $C_4$ eigenvalues at $\Gamma$ remain unchanged. Using Eq. (2), we identify PhCs with $d > 0$ to be topologically non-trivial, with bulk quadrupole invariant $q_{xy} = 1/2$, and PhCs with $d < 0$ to be topologically trivial with $q_{xy} = 0$.

A more-comprehensive topological phase diagram of our system is shown in Fig. 1f, determined by displacement $d$ and strength of gyromagnetic response $\kappa$. Both quadrupole topological insulators (green) and trivial insulators (orange) are identified, with the super-cell structure ($d = 0$) being the transition in between the two phases ($y$ axis). Interestingly, Chern insulating phases (purple) are observed at large displacements, consistent with the broken time-reversal symmetry[19,20].

**Nested Wilson loop and Wannier representation.** To confirm the quadrupole topology of our PhC, we explicit show that its dipole moments are zero ($p_x = p_y = 0$), but its quadrupole moment is non-zero ($q_{xy} = 1/2$). To this end, we perform two separate sets of calculations using two different methods: (1) the nested Wilson loop formulation[6]; (2) and the expectation value of the exponentiated quadrupole operator[21,22], and show they reach the same conclusions. Here, we present the nested Wilson loop calculations, leaving the second method in Supplementary Note 3. We start by computing the band structure and mode profiles ($E_z$)

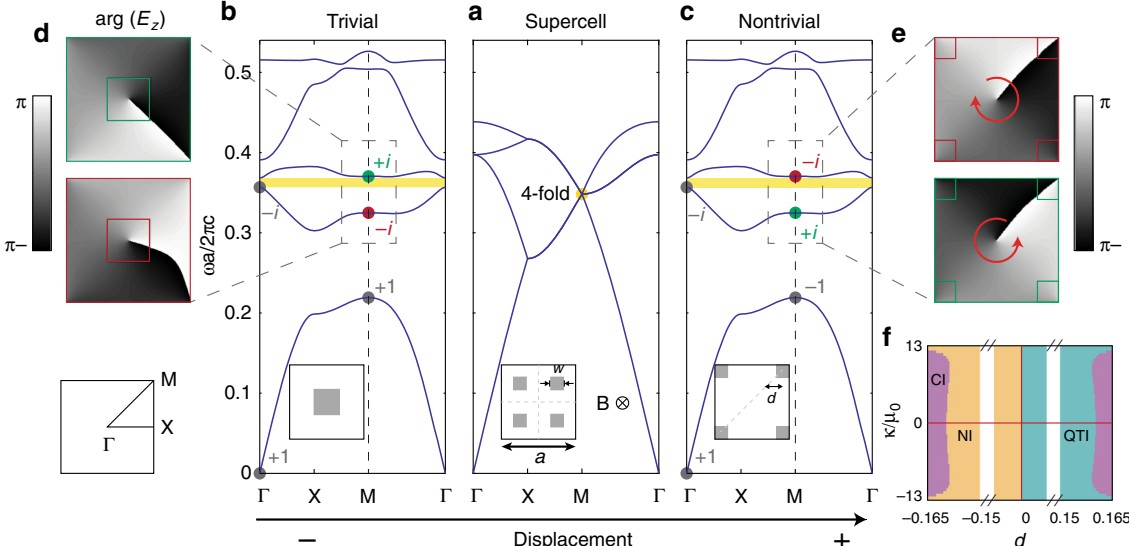

**Fig. 1 Quadrupole topological phase transitions through super-cell PhCs. a** Band degeneracies are created at the center (Γ) and corner (M) of the folded Brilluion zone for a 2 × 2 super-cell PhC (inset). An external magnetic field (B) induces gyromagnetic responses in YIG rods (gray squares), which breaks $T$ but preserves $C_4$. The gyromagnetic response $\kappa = 12.4\mu_0$ is used in the calculations. **b, c** As the rods simultaneously move inward or outward, the second gap is opened (shaded in yellow), but with different quadrupole phases: inward (outward) displacements with negative (positive) $d$ correspond to trivial (non-trivial) quadrupole phases. **d, e** $E_z$ mode profiles for the second and third modes at M feature a band inversion between the modes with $C_4$ index of $+i$ (green) and $-i$ (red) in the trivial and non-trivial phases. **f** Complete topological phase diagram of the PhCs, including quadrupole phases (green), trivial phases (orange), and Chern insulating phases (purple). Systems are gapless along the axes (red). NI normal insulator, QTI quadrupole insulator, CI Chern insulator.

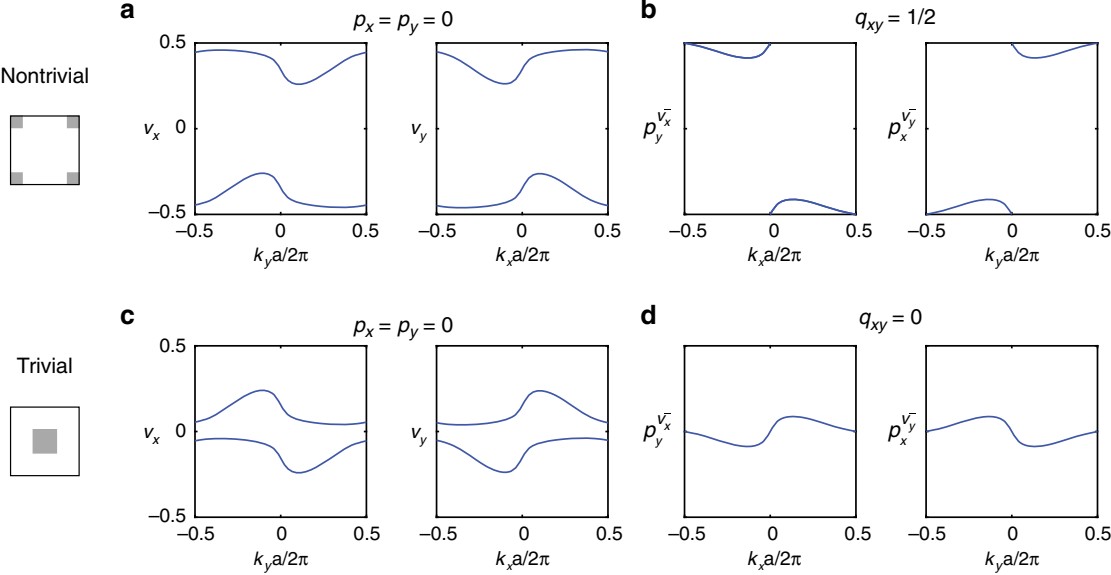

**Fig. 2 Confirmation of quadrupole and trivial PhCs through nested Wannier bands. a** Wannier bands, $\nu_x(k_y)$ and $\nu_y(k_x)$, are calculated for the first and second bands of the quadrupole PhC. Results show the bulk dipole momentum is zero ($p_x = p_y = 0$). **b** Calculations of the nested Wilson loops, $p_y^{\nu_x^-} = p_x^{\nu_y^-} = 0.5$, show the bulk quadrupole momentum $q_{xy}$ is non-trivial and quantized to 0.5. **c, d** Similar calculations repeated for a trivial PhC, showing zero bulk dipole moments ($p_x = p_y = 0$) and zero quadrupole moments ($p_y^{\nu_x^-} = p_x^{\nu_y^-} = 0$).

of the TM modes for a particular displacement of dielectric rods $d = a/4 - w/2$ (Fig. 2a, same as in Fig. 1c). Using these as input, we compute the Wannier bands $\nu_{x,y}$ for the two lowest-energy bands (Supplementary Note 1).

The results (Fig. 2a) show that the Wannier bands are related as $\{\nu_x^j(k_y)\} \to \{-\nu_x^j(-k_y)\}$ mod 1 owing to the presence of $C_2$ symmetry (Supplementary Note 2). Therefore, the bulk polarization $p_x$, which is simply the integral of the two Wannier bands over the 1D B.Z., vanishes: $p_x = \sum_{n=\pm} \int_{\text{B.Z.}} dk_y \, \nu_x^n(k_y) = 0$. A

similar argument can also be applied to bulk polarization $p_y$, proving the bulk dipole moments are zero: $p_x = p_y = 0$. Meanwhile, owing to the existence of a gap in the Wannier spectrum, the upper and lower Wannier bands can be separated into two sectors, labeled as $\nu_x^\pm$ ($\nu_y^\pm$), respectively. To confirm the quadrupole topology, we compute the polarizations of the Wannier bands within one sector, $p_y^{\nu_x^-}$ and $p_x^{\nu_y^-}$, using the nested Wilson loop formulation[6,23]. As shown in Fig. 2b, both Wannier band polarizations are quantized to be 1/2—due to the

simultaneous preservation of $M_x T$ and $M_y T$ symmetries (Supplementary Note 2)—which further confirms our PhC has a non-zero bulk quadrupole moment: $q_{xy} = 2p_x^{\nu_y^-} p_y^{\nu_x^-} = 1/2$.

To compare, we repeat the same set of calculations for a different unit cell with a negative displacement of $d = -(a/4 - w/2)$ (Fig. 2c). As shown, the Wannier bands are also gapped and sum to zero, meaning the bulk dipoles $p_{x,y}$ remain zero. However, the nested Wilson loops, $p_y^{\nu_x^-}$ and $p_x^{\nu_y^-}$, are also zero, leading to a trivial quadrupole moment $q_{xy} = 0$. This correspondence between positive (negative) displacements leading to non-trivial (trivial) quadrupole phases is consistent with our preceding conclusions based on $C_4$ symmetry eigenvalues. Importantly, the non-trivial and trivial PhCs in Fig. 2 can be simply related to each other, by shifting the choice of the unit cell center by $a/2$. This observation leads to an intuitive understanding of the difference in quadrupole moment between the two phases as discussed in Supplementary Note 5.

**Bulk-edge correspondence of Wannier bands.** Next, we present the physical consequences of quadrupole topological PhCs at interfaces, originating from the bulk-edge correspondence of the Wannier bands. Following classical electromagnetism, a non-zero bulk quadrupole moment in a finite system is manifested as edge polarizations at its boundaries. Here, we study the 1D interfaces between quadrupole (trivial) PhCs and perfect electric conductors (PECs). We find that the non-trivial quadrupole topology indeed leads to a quantized edge polarization along the interface, which is absent for a trivial PhC. Specifically, we consider a strip of 20 unit cells of the quadrupole (trivial) PhC design, which satisfies periodic boundary condition in the $x$-direction and closed boundary condition in the $y$-direction, owing to the two PECs on top and bottom as shown in Fig. 3a, b. The energy dispersions $\omega(k_x)$ of the quadrupole and trivial strips (Fig. 3c, d) share the same bulk energy spectra (gray areas) but have different edge dispersions (gray solid lines), owing to their different edge terminations. The Wannier centers $\nu_x$ are calculated for the two different strips based on their energy dispersions and Bloch mode profiles (Fig. 3e, f). For the topological strip, two additional Wannier states (red circles) are found to emerge outside the Wannier bands (blue) at the middle of the gap ($\pm 0.5$), which is protected by the non-trivial bulk quadrupole moment. This can be understood in a similar way as the in-gap edge states in the 1D SSH model with non-trivial bulk polarization. The quantization of the mid-gap Wannier states at $\pm 0.5$ is due to the additional $M_x T$ symmetry in our system, which is retained even at the boundary of a finite strip as shown in Fig. 3a. (Supplementary Note 2). In comparison, no additional Wannier states are found inside the Wannier gaps for the trivial strip, consistent with the lack of bulk quadrupole moment.

Furthermore, the two mid-gap Wannier states in the topological strip are spatially localized at the top and bottom edges. To demonstrate this, we further study the spatial distribution of Wannier states by calculating the polarization density $p_x$ as a function of position along $y$ (Supplementary Note 6). In order to choose a definite sign of the polarization, we introduce an infinitesimal perturbation to break the $C_2$ symmetry of the semi-infinite strip. For the quadrupole PhC, as shown in Fig. 3g, there are non-zero polarization densities developed near the two edges at $y = 0$ and $y = 20a$, and the edge polarizations are quantized to $\pm 0.5$. In comparison, neither edge nor bulk polarization are observed in the trivial PhC (Fig. 3h).

**Fractional corner charges and filling anomaly.** Finally, we show the physical consequence of quadrupole PhCs as localized 0D corner states, which are the photonic analogs of states responsible for filling anomalies and fractional charges in an electronic setting[24]. To this end, we solve the eigenstates in a finite 2D quadrupole PhC enclosed by PECs, with a thin air gap in between (Fig. 4a). The eigenstates are labeled according to their energies, with the lowest-energy state labeled as 1. Aside from delocalized

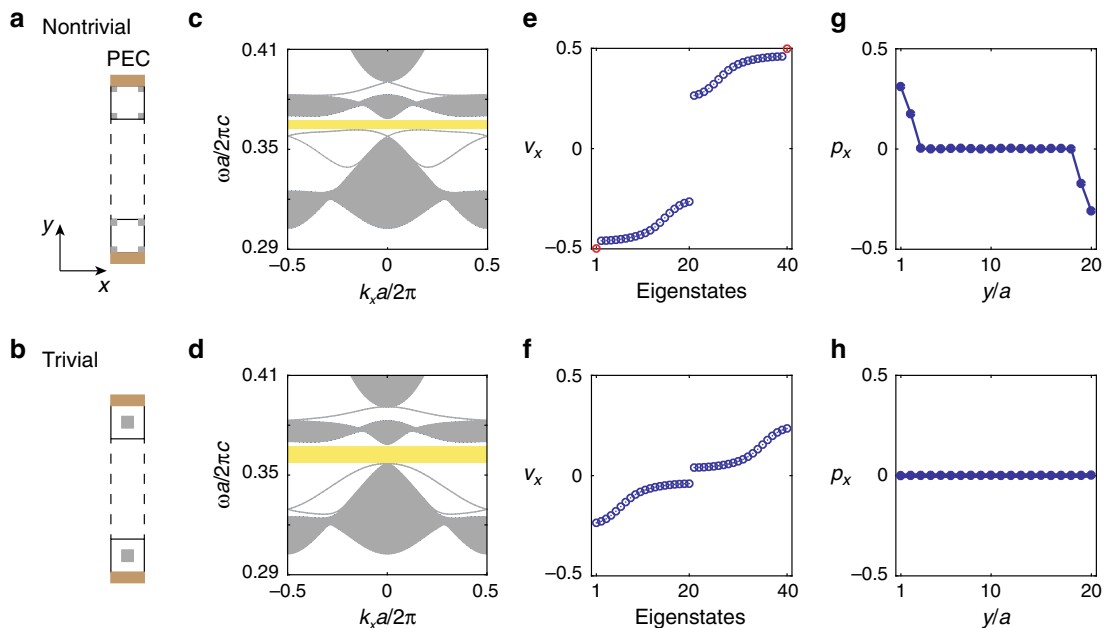

**Fig. 3 Physical consequences of quadrupole topological PhCs at 1D interfaces. a, b** Two 1D interfaces are created between a strip of quadrupole (trivial) PhC (same as Fig. 1b,c) and perfect electric conductors (PECs). **c, d** Energy dispersions of the two strip setups. **e** For the non-trivial setup, Wannier centers of the eigenmodes show two Wannier states at ±0.5, which are outside the bulk Wannier bands owing to the non-trivial bulk quadrupole moment. **f** In comparison, no in-gap Wannier states are observed in the trivial setup. **g** Calculation of the spatial density of polarization $p_x(y)$ shows the in-gap Wannier states are localized at the top ($y = 20a$) and bottom interfaces ($y = 0$). **h** Similar calculations repeated for the trivial PhC, showing no edge polarization, which is consistent with the trivial quadrupole moment.

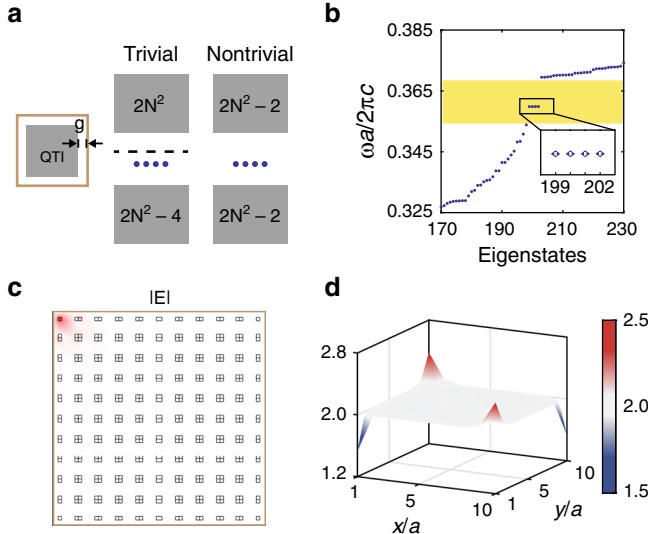

**Fig. 4 Topologically protected corner states and filling anomaly in a 2D system. a** For a finite PhC with N×N unit cells enclosed by PECs, the counting of the number of eigenstates below and above an energy gap are distinct between quadrupole and trivial phases. **b** Eigenstates of a 10 × 10 quadrupole PhC, showing four degenerate corner states inside the bulk gap, shaded in yellow. **c** Mode profile ($|E_z|$) of one of the corner states. **d** Spatial distribution of the accumulative electromagnetic energy density for the lowest 200 energy eigenstates in the quadrupole PhC, showing fractional occupations ($2 \pm 0.5$) at the corners.

bulk states, four degenerate states—199 through 202—are found, with their energies inside the bulk energy gap (Fig. 4b). Their mode profiles ($E_z$) confirm that these states are spatially localized at the four corners, with one example shown in Fig. 4c. Owing to the lack of chiral symmetry in our system, which is generic in the continuum theory expanded around non-zero frequencies, the energy of the four corner states is not pinned at the center of the bulk energy gap[25]; instead they can be shifted, even immersed into the bulk continuum, by modifying the refractive index at the corners. This renders the appearance of corner states less of an essential signature of quadrupole phases[10,11]. In fact, corner states are also found in other higher-order topological phases with vanishing bulk quadrupole moments[23–30].

Instead, here we illustrate the non-trivial quadrupole nature of our PhC using the filling anomaly, by counting the number of energy eigenstates below and above a given bulk energy gap[10] (Fig. 4a). Specifically, even though trivial samples may support corner states, they originate from either the top or bottom band alone, leaving $2N^2-4n$ states in the bulk continuum ($n$ is an integer). On the other hand, for non-trivial PhCs, the number of states below and above the energy gap are both $2N^2 - 2 = 198$ ($N = 10$ in our case). This indicates the four degenerate corner states we observed are "contributed" by both the top and bottom bands together, and thus proves the non-trivial quadrupole topology of our design. As a consequence, for a quadrupole PhC, quantized fractional charges appear at four corners of a finite sized system (Fig. 4d) when calculate the spatial distribution of the lowest 200 energy states, in a similar vein as the calculations of charge density in electronic systems at "half-filling" (here, we have introduced an infinitesimal perturbation to break $C_4$ symmetry in order to split the fourfold degenerate corner states). Remarkably, these corner charges are shared by two convergent dipoles on the two perpendicular edges, as the magnitude of the corner charges is equal to the edge polarizations. This further confirms that these corner charges originate from a non-zero bulk

quadrupole moment. We point out that the observed fractional corner charges arise from the fundamental difference in the counting of bulk states[10], and was recently proposed in electronic systems by Benalcazar et al.[24] as a filling anomaly: a mismatch between number of states in an energy band and the number of electrons required for charge neutrality.

## Discussion

In summary, we present quadrupole topological photonic crystals with truly quantized invariants and the physical consequences at material's edges and corners. The proposed gyromagnetic PhCs can be readily realized in the microwave regime. Meanwhile, the coexistence of multiple topological phases in our system, both quadrupole TIs and Chern insulators, provides a versatile platform to further demonstrate topological photonic circuits with protected elements immune to disorders in various dimensions. Finally, our findings of inducing quadrupole phase transitions and quantizing quadrupole moments—through crystalline symmetries in conjunction with broken time-reversal symmetry—can also be applied to other wave systems, including electrons, phonons, and polaritons.

## Methods

**Numerical simulation of Maxwell's equation using the finite element method**. The band structures and mode profiles are calculated using the Finite Element Method in COMSOL Multiphysics 5.4. Specifically, we compute the band structures and mode profiles in a 2D geometry with periodic boundary conditions along all directions. The corresponding Wannier bands are calculated by using the Bloch mode profiles as input (Supplementary Note 1).

## Data availability

The data that support the findings of this study are available from the corresponding author upon reasonable request.

## Code availability

The code that supports the plots within this paper and other findings of this study is available from the corresponding authors upon reasonable request.

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

## Acknowledgements
This work was partly supported by the National Science Foundation through the University of Pennsylvania Materials Research Science and Engineering Center DMR-1720530 and the US Office of Naval Research (ONR) Multidisciplinary University Research Initiative (MURI) grant N00014-20-1-2325 on Robust Photonic Materials with High-Order Topological Protection. L.H. was supported by the Air Force Office of Scientific Research under award number FA9550-18-1-0133. Work by Z.A. and E.M. on the symmetry analysis of the quadrupole phase was supported by the Department of Energy, Office of Basic Energy Sciences under grant DE FG02 84ER45118. B.Z. was supported by the Army Research Office under award contract W911NF-19-1-0087.

## Author contributions
L.H. and B.Z. conceived the idea. L.H. carried out numerical simulations. L.H., Z.A., E.M., and B.Z. discussed and interpreted the results. L.H and B.Z. wrote the paper with contribution from all authors. B.Z. supervised the project.

## Competing interests
The authors declare no competing interests.
