## [Peer Review file · Nature Communications]

Reviewers' comments:

Reviewer #1 (Remarks to the Author):

The authors present a theoretical study of the emergence of quadrupole topological phases in a two-dimensional photonic crystal. They basically take the concepts developed by Benalcazar and coworkers in Science 357, 61–66 (2017) and PRB 96, 245115 (2017) to characterise quadrupole phases, and adapt them to the case of a photonic crystal characterised by a spatially patterned dielectric constant and gyromagnetism.

I think this work is important because it translates quadrupole topological insulators, mainly described so far by tight-binding models, to a photonic crystal context in a direct way. However, the employed methods and results present many parallels with the above mentioned papers, and in this sense I am not sure they provide genuinely new physical insights. At least the present version does not emphasize enough the newer conceptual elements, if any, beyond the adaptation of those methods to a continuous system and the proposal of a specific platform. For this reason, I am not sure this paper makes a case for publication in a general audience journal such as Nature Communications.

Beyond this general remark, I have a few minor comments:

- All calculations seem to have been done in 2D. Is this right?
- In page 3, the method used to calculate the bands shown in Fig. 1 should be mentioned. Are these finite elements simulations of Maxwell's equations?
- Page 5, the discussion about gapped Wannier bands is a bit confusing, in particular the sentence: "Meanwhile, as the two Wannier bands are gapped — one between -0.5 and 0 and the other between 0 and 0.5 — the Wannier states can be separated into two sectors, labeled as $v \pm x$ ($v \pm y$)." Do you refer simply to the upper and lower bands in panels 2a?
- Page 6. Where is it shown that Wannier bands sum to zero? What do you mean?
- Is there an intuitive way of connecting Figs 3c and 3e?

Reviewer #2 (Remarks to the Author):

The manuscript "Quadrupole Topological Photonic Crystals" by Li He et al. presents a study on the quadrupolar topological states in photonic crystals (PhC).

More in detail, the authors analyze how emerge quadrupolar topological modes in a 2D PhC made of square pillars of gyromagnetic rods (Yttrium Iron Garnet) in air, after applying a magnetic field in order to break time-reversal symmetry. In particular, they show quantization of the quadrupolar moment purely by crystalline symmetries using different approaches. As well as, they highlight the boundary manifestations of non-trivial quadrupole phases as quantized polarizations at edges and bound states at corners.

High order topological phase is extremely interesting and timely, in particular in the photonic world where seems simpler to build and 'use' their rich phenomenology. The manuscript is clear and the figures are crafted. In other words, I think that the manuscript presents definitely important

contributions to the field, anyway I have some doubt on the methodology and the novelty that will good to dissipate.

1. I have unclear if the authors claim that the crystalline symmetry only quantize the quadrupole moment, or is also responsible for the existence of protected edge states. I'm asking it because if we have an edge we break the crystal symmetry.

2. It is not clear from the manuscript how bands are calculated, please give details. This is quite important point as the authors highlight. Seems that a full Maxwell problem is solved without any approximation, is that right? For example, in photonics, approximations as the tight-binding method could lead to the wrong conclusion due to the long-range interaction between nanoparticles (see for example ACS Photonics 2018, 5, 2271–2279).

3. My bigger doubt is about the novelty of this work and why should be published here. For instance in photonics similar argument and results have been analyzed in ref 26 or in the following two references

a. Nature Photonics volume 14, pages 89–94(2020)

b. arXiv:1912.06517, Twisted quadrupole topological photonic crystal

I encourage the authors to make a solid case.

Reviewer #3 (Remarks to the Author):

The authors investigate quadrupole topological insulators arising due to broken time-reversal symmetry in the presence of protecting crystalline symmetries. They characterize the topology by means of Wilson loops, symmetry eigenvalues and a filling anomaly. They demonstrate these concepts on an example lattice that undergoes a phase transition as geometric parameters are modified.

I consider the paper interesting, well written and concise and therefore enthusiastically recommend publication in Nature Communications. I found particularly interesting the derivation of a topological invariant for quadrupole systems based on symmetry eigenvalues at high-symmetry points, the fact that the authors derive topological invariants that are valid outside a tight-binding approximation and the discussion of filling anomaly vs edge states in the discussion of the experimental signatures of quadrupole topology.

Reviewer #1 (Remarks to the Author):

The authors present a theoretical study of the emergence of quadrupole topological phases in a two-dimensional photonic crystal. They basically take the concepts developed by Benalcazar and coworkers in Science 357, 61–66 (2017) and PRB 96, 245115 (2017) to characterise quadrupole phases, and adapt them to the case of a photonic crystal characterised by a spatially patterned dielectric constant and gyromagnetism.

I think this work is important because it translates quadrupole topological insulators, mainly described so far by tight-binding models, to a photonic crystal context in a direct way.

Reply

We thank reviewer 1 for the positive comments on our manuscript.

However, the employed methods and results present many parallels with the above mentioned papers, and in this sense I am not sure they provide genuinely new physical insights. At least the present version does not emphasize enough the newer conceptual elements, if any, beyond the adaptation of those methods to a continuous system and the proposal of a specific platform. For this reason, I am not sure this paper makes a case for publication in a general audience journal such as Nature Communications.

Beyond this general remark, I have a few minor comments:

Reply

We agree with the reviewer that the essential concept of topological multipole insulators was first formulated in the seminal papers by Benalcazar and coworkers through analyzing minimal lattice models in *electronic* systems. Recently, there has been great effort to demonstrate the quadrupole phases in classical wave systems (photons, phonons, etc.), including the following works:

1. Peterson et al., Nature 555, 346 (2018)
2. Serra-Garcia et al., Nature 555, 342 (2018)
3. Imhof et al., Nature Physics 14, 925 (2018)
4. Mittal et al., Nature Photonics 1 (2019).

Compared to these studies, we would argue that our work provides unique physical insights for topological multipole insulators, which are critical for their future realizations in realistic materials and device-level applications.

First, despite the various platforms adopted, the current theoretical framework – based on tight-binding models – limits the experimental systems to coupled-resonator arrays. However, as also pointed out by reviewer 2, most realistic systems with subwavelength features, such as photonic crystals, cannot be modeled *discretely* and instead require a new theoretical framework. For this reason, we expand the theory of topological multipole insulators to *continuous* Maxwell's equations, for the first time, which is nontrivial in its own right.

Second, we reveal the crucial role of *time-reversal symmetry* in topological quadrupole phases, which has been overlooked before. The introduction of flux threading with positive/negative coupling in all previous studies preserves time-reversal symmetry, yet it leads to *non-quantized* bulk multipole moments in the continuum limit (see for example arXiv:1911:06068). How can one quantize quadrupole moments q_{xy} in continuous systems remains elusive until we show, in this work, q_{xy} can be

quantized by the simultaneous presence of crystalline symmetry and **broken** time-reversal symmetry. Our work provides the ground to study symmetry-protected quadrupole phases, which can provide further insights to even-higher-order topological phases.

However, we agree with the reviewer that our manuscript, in its past version, did not adequately emphasize our novelties. To better highlight the significance, we have now modified the introduction part of the main text as following:

“Recently, quadrupole TIs have been demonstrated in a number of classical systems, ranging from microwave and optics to acoustics and circuits. Most experiments are analyzed as lattice models, following the tight-binding approximation. In these lattice models, the adopted symmetry required to quantize the quadrupole moment is reflection or four-fold rotation with threaded π -fluxes, and hence these systems are time-reversal invariant. Unfortunately, most realistic systems with subwavelength features, such as photonic crystals, cannot be modeled discretely but require a different continuum approach. For example, the practical implementations of many lattice models no longer preserve quantized bulk quadrupole moments in the continuum theory. Additionally, most experimental works solely used the existence of in-gap corner states as the measure of bulk quadrupole topology. The validity relies on the presence of additional chiral symmetries in the lattice model, which is often not preserved in the continuum theory.

Here we find solutions to continuous Maxwell’s equations in gyromagnetic photonic crystals that are the electrodynamic analogs to quadrupole topological phases. The proposed topological PhCs have quantized bulk quadrupole moments, which are protected by the simultaneous presence of crystalline symmetries and broken time-reversal symmetry. In particular, we show it is essential to break time-reversal symmetry – to open the energy gap – while preserving crystalline symmetries – to quantize bulk dipole and quadrupole moments. In addition, we show the existence of corner states is neither sufficient nor necessary condition for quadrupole phases. Instead, we validate the bulk quadrupole nature through analyzing the Wannier-band polarization and its manifestations at boundaries as quantized edge polarizations and fractional corner charges. All calculations are based on realistic parameters that are readily available in the microwave regime.

Comment 1

All calculations seem to have been done in 2D. Is this right?

Reply 1

Yes, all calculations have been performed in 2D as we consider an infinite photonic crystal along the out-of-plane (z) direction, or a finite-thickness photonic crystal sandwiched between two metallic plates as in experiments (Wang *et al.* Nature 2009).

To clarify this issue, we have now added the following sentence on page 3:

“We start by presenting the topological phase transition between a trivial (Fig. 1a, left panel) and a quadrupole two-dimensional (2D) PhC (right panel). The 2D PhC consists of gyromagnetic rods in air and is homogeneous along the out-of-plane (z) direction. Experimentally, this boundary condition can also be realized using metals.”

Comment 2

In page 3, the method used to calculate the bands shown in Fig. 1 should be mentioned. Are these finite elements simulations of Maxwell's equations?

Reply 2

Yes, the band structures shown in Fig. 1 are obtained by directly solving Maxwell's equations using the Finite Element Method (COMSOL Multiphysics).

To further clarify this point, we have now added detailed descriptions of band structure calculation in the Method Section:

“Numerical simulation of Maxwell's equation using the Finite Element Method. The band structures and mode profiles are calculated using the Finite Element Method in COMSOL Multiphysics 5.4. Specifically, we compute the band structures and mode profiles in a 2D geometry with periodic boundary conditions along all directions. The corresponding Wannier bands are calculated by using the Bloch mode profiles as input (Supplementary Information 1).”

Comment 3

Page 5, the discussion about gapped Wannier bands is a bit confusing, in particular the sentence: “Meanwhile, as the two Wannier bands are gapped — one between -0.5 and 0 and the other between 0 and 0.5 — the Wannier states can be separated into two sectors, labeled as $v_{\pm x}$ ($v_{\pm y}$).” Do you refer simply to the upper and lower bands in panels 2a?

Reply 3

Yes, the gapped Wannier bands refer to the upper and lower bands in Fig. 2a.

To avoid possible confusion, we have now modified our description as following:

“Meanwhile, due to the existence of a gap in the Wannier spectrum, the upper and lower Wannier bands can be separated, labeled as v_x^{\pm} (v_y^{\pm}), respectively.”

Comment 4

Page 6. Where is it shown that Wannier bands sum to zero? What do you mean?

Reply 4

Here, by “the sum of Wannier bands”, we are referring to the polarization p_x , defined as the integral of the two Wannier bands over k_y :

$$p_x = \sum_{n=\pm} \int_{BZ} dk_y v_x^n(k_y).$$

As the two Wannier bands (Fig. 2a) are related through C_2 operation:

$$\{v_x^{\pm}(k_y)\} \xrightarrow{C_2} \{-v_x^{\pm}(-k_y)\} \text{ mod } 1,$$

the bulk polarization vanishes $p_x = 0$. Therefore, “the Wannier bands sum to zero.” Similarly, bulk polarization p_y is also 0 (Fig. 2a right panel).

To avoid potential confusion, we have now added the following sentence on page 5:
“The results (Fig. 2a, left panel) show that two Wannier bands are related as $\{v_x^\pm(k_y)\} \rightarrow \{-v_x^\pm(-k_y)\} \bmod 1$ due to the presence of C_2 symmetry (Supplementary Information 2). Therefore, the bulk polarization p_x , which is simply the integral of the two Wannier bands over the 1D B.Z., vanishes: $p_x = \sum_{n=\pm} \int_{BZ} dk_y v_x^n(k_y) = 0$. A similar argument can also be applied to bulk polarization p_y , proving the bulk dipole moments are zero: $p_x = p_y = 0$.”

Comment 5

Is there an intuitive way of connecting Figs 3c and 3e?

Reply 5

A possible intuitive connection between edge bands in Fig. 3c and in-gap Wannier states in Fig. 3e may be the follows: first, one can potentially completely separate the (two) edge bands in Fig. 3c from the bulk continuum by tuning the boundary details without breaking bulk symmetries. Then, the two Wannier states, constructed from these edge bands, must be localized on the top and bottom edges. The Wannier centers of these edge bands (or the polarizations of the edge bands) are quantized to either 0 or 1/2, due to $M_x T$ symmetry, which are exactly the in-gap Wannier states in Fig. 3e.

Reviewer #2 (Remarks to the Author):

The manuscript "Quadrupole Topological Photonic Crystals" by Li He et al. presents a study on the quadrupolar topological states in photonic crystals (PhC).

More in detail, the authors analyze how emerge quadrupolar topological modes in a 2D PhC made of square pillars of gyromagnetic rods (Yttrium Iron Garnet) in air, after applying a magnetic field in order to break time-reversal symmetry. In particular, they show quantization of the quadrupolar moment purely by crystalline symmetries using different approaches. As well as, they highlight the boundary manifestations of non-trivial quadrupole phases as quantized polarizations at edges and bound states at corners.

High order topological phase is extremely interesting and timely, in particular in the photonic world where seems simpler to build and 'use' their rich phenomenology. The manuscript is clear and the figures are crafted. In other words, I think that the manuscript presents definitely important contributions to the field,

Reply

We thank reviewer 2 for the positive comments on our manuscript.

anyway I have some doubt on the methodology and the novelty that will good to dissipate.

Comment 1

I have unclear if the authors claim that the crystalline symmetry only quantize the quadrupole moment, or is also responsible for the existence of protected edge states. I'm asking it because if we have an

edge we break the crystal symmetry.

Reply 1

For our photonic crystal design, certain bulk crystalline symmetries are preserved both in the bulk and on (certain) edges. For example, $M_x T$, the combined operation of mirror reflection along x and time reversal, is preserved on the edges shown in Fig. 3a. Under this operation, Wannier centers transform as $v_x \rightarrow -v_x \text{ mod } 1$. As a result, our edge Wannier states are quantized: $v_x = \pm 0.5$.

To clarify this point, we have now added a sentence on page 7.

“The quantization of the mid-gap Wannier states at ± 0.5 is due to the additional $M_x T$ symmetry in our system, which is retained even at the boundary of a finite strip as shown in Fig. 3a.”

Comment 2

It is not clear from the manuscript how bands are calculated, please give details. This is quite important point as the authors highlight. Seems that a full Maxwell problem is solved without any approximation, is that right? For example, in photonics, approximations as the tight-binding method could lead to the wrong conclusion due to the long-range interaction between nanoparticles (see for example ACS Photonics 2018, 5, 2271–2279).

Reply 2

Yes, the photonic bands shown in Fig. 1 are obtained by solving the full Maxwell’s equation using Finite Element Method.

To clarify any potential confusions, we have now provided detailed description of band calculation in the Method Section.

“Numerical simulation of Maxwell’s equation using the Finite Element Method. The band structures and mode profiles are calculated using the Finite Element Method in COMSOL Multiphysics 5.4. Specifically, we compute the band structures and mode profiles in a 2D geometry with periodic boundary conditions along all directions. The corresponding Wannier bands are calculated by using the Bloch mode profiles as input (Supplementary Information 1).”

Comment 3

My bigger doubt is about the novelty of this work and why should be published here. For instance in photonics similar argument and results have been analyzed in ref 26 or in the following two references

- a. Nature Photonics volume 14, pages 89–94(2020)
- b. arXiv:1912.06517, Twisted quadrupole topological photonic crystal

I encourage the authors to make a solid case.

Reply 3

We respectively disagree with the reviewer, in that our quadrupole photonic crystals belong to a new type of topological crystalline insulators compared to the above-mentioned works. They are enabled by distinct crystalline symmetries and have different boundary manifestations.

Specifically, the works presented in ref. 26 and Nature Photonics vol 14 reported high order topological phases realized in photonic and phononic systems. The topological nature of their system is characterized by dipole moments (or bulk polarization) rather than higher multipole moments. In contrast, our work extends the discussion from dipole moments to higher multipole moments, like the quadrupole, and demonstrates their quantization in the presence of crystalline symmetries.

The work (arXiv:1912.06517) was posted on arXiv in December 2019, a month later than our submission. In this work, they studied *anomalous quadrupole topology* classified by nonsymmorphic symmetries, which differs fundamentally from the quadrupole phases as presented in our work.

In addition, in all above-mentioned works, the bulk nontrivial topologies are identified by measuring the corner states within the energy band gap. The validity holds only when there is additional chiral symmetry in the energy spectrum. However, through numerical calculations, we show the bulk quadrupole topology is manifested as filling anomalies and fractional corner charges. To the best of our knowledge, none of these observations have been made in any photonic systems yet.

Reviewer #3 (Remarks to the Author):

The authors investigate quadrupole topological insulators arising due to broken time-reversal symmetry in the presence of protecting crystalline symmetries. They characterize the topology by means of Wilson loops, symmetry eigenvalues and a filling anomaly. They demonstrate these concepts on an example lattice that undergoes a phase transition as geometric parameters are modified.

I consider the paper interesting, well written and concise and therefore enthusiastically recommend publication in Nature Communications. I found particularly interesting the derivation of a topological invariant for quadrupole systems based on symmetry eigenvalues at high-symmetry points, the fact that the authors derive topological invariants that are valid outside a tight-binding approximation and the discussion of filling anomaly vs edge states in the discussion of the experimental signatures of quadrupole topology.

Reply

We thank reviewer #3 for the positive comments on our manuscript.

REVIEWERS' COMMENTS:

Reviewer #1 (Remarks to the Author):

I find the revised version of the manuscript clearer. In particular, the novelty and relevance of this work is now much clearer. The authors have diligently modified the manuscript following the suggestions and questions of the referees. I thus recommend publication in Nature Communications.

I just have a minor question regarding the time reversal symmetry breaking. I guess the magnitude of the yellow gap visible in Figs. 1, 3 and 4 is mainly given by the gyromagnetic response κ . The deformation alone and the gyromagnetic response alone (Fig. 1a "supercell" panel) do not open a gap. Is that correct?

Reviewer #2 (Remarks to the Author):

I think that the manuscript has been definitely improved, and the obscure section on the EM calculation has been elucidated. I am happy to see this manuscript published as it is.

Reviewer #1 (Remarks to the Author):

I find the revised version of the manuscript clearer. In particular, the novelty and relevance of this work is now much clearer. The authors have diligently modified the manuscript following the suggestions and questions of the referees. I thus recommend publication in Nature Communications.

Reply

We thank reviewer 1 for the positive comments on our revised manuscript.

I just have a minor question regarding the time reversal symmetry breaking. I guess the magnitude of the yellow gap visible in Figs. 1, 3 and 4 is mainly given by the gyromagnetic response κ . The deformation alone and the gyromagnetic response alone (Fig. 1a “supercell” panel) do not open a gap. Is that correct?

Reply

Yes, the energy gap in Fig. 1,3,4 is a direct consequence of time-reversal symmetry breaking. Therefore, in the weak perturbative regime, the energy gap is proportional to T-breaking strength (or the magnitude of gyromagnetic response). On the other hand, the degeneracies of the “supercell” dispersion in Fig. 1a is due to band folding, and hence can only be gapped out by deforming the rods.

To clarify this point, we have now added a remark on page 4.

“On either side of the transition point, the band structure is fully gapped due to T-breaking (shaded in yellow) and supports...”

Reviewer #2 (Remarks to the Author):

I think that the manuscript has been definitely improved, and the obscure section on the EM calculation has been elucidated. I am happy to see this manuscript published as it is.

Reply

We thank reviewer 2 for the positive comments on our revised manuscript.